# Many-body quantum dynamics slows down at low density

**Xiao Chen[1], Yingfei Gu[2] and Andrew Lucas[3⋆]**

**1** Department of Physics, Boston College, Chestnut Hill, MA 02467, USA
**2** Department of Physics, Harvard University, Cambridge MA 02138, USA
**3** Department of Physics and Center for Theory of Quantum Matter,
University of Colorado, Boulder CO 80309, USA

⋆ andrew.j.lucas@colorado.edu

## Abstract

We study quantum many-body systems with a global U(1) conservation law, focusing on a theory of $N$ interacting fermions with charge conservation, or $N$ interacting spins with one conserved component of total spin. We define an effective operator size at finite chemical potential through suitably regularized out-of-time-ordered correlation functions. The growth rate of this density-dependent operator size vanishes algebraically with charge density; hence we obtain new bounds on Lyapunov exponents and butterfly velocities in charged systems at a given density, which are parametrically stronger than any Lieb-Robinson bound. We argue that the density dependence of our bound on the Lyapunov exponent is saturated in the charged Sachdev-Ye-Kitaev model. We also study random automaton quantum circuits and Brownian Sachdev-Ye-Kitaev models, each of which exhibit a different density dependence for the Lyapunov exponent, and explain the discrepancy. We propose that our results are a cartoon for understanding Planckian-limited energy-conserving dynamics at finite temperature.

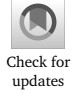

# 1   Introduction

There is a conjectured universal "bound on chaos" [1] in many-body quantum systems: loosely speaking, a suitably defined out-of-time-ordered correlator (OTOC) at finite temperature is constrained to obey

$$\text{tr}\left(\sqrt{\rho}[A(t),B]^{\dagger}\sqrt{\rho}[A(t),B]\right) \lesssim \frac{1}{N}e^{\lambda_{\text{L}}t}, \tag{1.1}$$

at sufficiently small $t > 0$, with $\rho \sim \exp[-H/T]$ the thermal density matrix at temperature $T$, and with Lyapunov exponent

$$\lambda_{\text{L}} \leq 2\pi T. \tag{1.2}$$

(We work in units with $\hbar = k_{\text{B}} = 1$.) Originally, this rather abstract inequality was motivated by observations about quantum gravity [2]; indeed, saturation of (1.2) is believed to be achieved only by gravitational theories (described by many-body systems, in accordance with the holographic principle). However, from at least a heuristic perspective, this inequality is also sensible physically: at low temperature, the dynamics is restricted to increasingly few thermally activated degrees of freedom. The dynamics must necessarily slow down accordingly, and (1.2) (ignoring the precise $2\pi$ prefactor) is simply fixed by the Heisenberg uncertainty principle:

$$\hbar \lesssim \Delta E \Delta t \sim \frac{T}{\lambda_{\text{L}}}. \tag{1.3}$$

This is one manifestation of a conjectured "Planckian" bound on quantum dynamics and thermalization, whereby the fastest time scale (at least, of thermalization) in a low temperature quantum system is $1/T$. Heuristic evidence for quantum dynamics being limited by the time scale $\hbar/k_{\text{B}}T$ has arisen in many fields ranging from holographic field theories [3,4] to quantum critical theories [5,6], strongly correlated electrons [7–10] and quark-gluon plasma [11].

Of course, the argument (1.3) is far from rigorous, and strictly speaking there are plenty of counter-examples to (1.1), e.g. in free fermion models [1,12]. Is it possible, at least under certain circumstances, to prove that quantum dynamics truly must slow down at low energy? More broadly, can we show unambiguously that quantum dynamics has to slow down in any kind of constrained subspace? While this might seem intuitive, and there is certainly evidence for this [13–15], proving such a statement has been notoriously challenging, and very few rigorous results are known. The standard approach for constraining quantum dynamics is based on the Lieb-Robinson theorem [16], which applies to operator norms and holds in every

state. By construction, therefore, Lieb-Robinson bounds are not useful at finding temperature-dependent bounds on quantum dynamics [17]. While recently these techniques have been improved to obtain temperature-dependent bounds on the velocity of information scrambling in one dimensional models [15], the resulting bounds depend on multiple microscopic model details.

It is almost certain that a rigorous derivation of (1.2), even in restricted models, is quite challenging without physical assumptions about thermalization. Clearly, a qualitatively important feature of low temperature dynamics is that it is restricted to low energy states in the Hilbert space. In this paper, we elect to study a simpler way to restrict dynamics to exponentially small parts of Hilbert space. Rather than cooling a system down to low temperature, we elect to study a system with a conserved U(1) charge, and in a highly polarized state with very low charge density $\overline{n} \ll 1$; here $\overline{n}$ denotes the probability that any lattice site is occupied. Elementary combinatorics demonstrates that exponentially small fractions of quantum states have $\overline{n} \ll 1$, even at infinite temperature. We will show explicitly how these constraints on accessible states qualitatively modify bounds on quantum dynamics and OTOC growth. In Section 3, we will show that in models where a Lyapunov exponent is well-defined,

$$\lambda_{\mathrm{L}} \le \lambda_* \overline{n}^{\gamma}, \tag{1.4}$$

where the exponent $\gamma > 0$ depends on basic details about the model (number of terms in interactions). There is a universal bound

$$\gamma \ge \frac{1}{2}, \tag{1.5}$$

valid for every theory; however, in certain cases, we can do parametrically better. (1.4) implies that in *every* theory with a U(1) conservation law (and a discrete Hilbert space), at least some kinds of quantum scrambling always become parametrically slow. The way which we derive this result is inspired by [12], which conjectured a similar phenomenon for energy conserving dynamics at finite temperature. However, our work will be more precise.

To understand the origin of the generic bound (1.5), let us recall that the growth of out-of-time-ordered correlators comes from the "size" of operators increasing (see Section 2 for details). Consider for simplicity a model of fermions, with creation and annihilation operators $c_i^{\dagger}$ and $c_i$. The simplest possible growth mechanism for a time evolving annihilation operator is (schematically)

$$c_i(t) = c_i + \sum_{j,k,l} J_{ijkl} c_j^{\dagger} c_k c_l t + \cdots . \tag{1.6}$$

Now, at low density, the second term above can only survive an expectation value if $c_j^{\dagger}$ acts on a state where site $j$ is occupied. The fraction of states in the thermal ensemble where this site is occupied is $\overline{n}$. So one might naively expect $\lambda_{\mathrm{L}} \sim \overline{n}$ is the fastest possible operator growth, since each power of $t$ will come with a factor of $\overline{n}$ (we can only add $c_j^{\dagger}$ and $c_k$ in "pairs", by charge conservation). However, since OTOCs such as (1.1) contain *two* commutators, the leading order density-dependent contribution to the OTOC will be

$$\frac{1}{\mathrm{tr}(\sqrt{\rho} c_i^{\dagger} \sqrt{\rho} c_i)} \mathrm{tr} \left( \sqrt{\rho} \sum_{j,k,l} J_{ijkl} c_l^{\dagger} c_k^{\dagger} c_j t \times \sqrt{\rho} \sum_{j,k,l} J_{ijkl} c_j^{\dagger} c_k c_l t \right) \sim \overline{n} t^2 . \tag{1.7}$$

This suggests that (1.5) is actually the optimal bound on OTOC growth. We will show that this argument is correct. In particular, in Section 4, we find that the $\overline{n}$ dependence of (1.4) is saturated by the charged Sachdev-Ye-Kitaev (SYK) model [18, 19]. Thus, (1.4) cannot be parametrically improved.

At the same time, we will see that the density dependence of (1.4) is qualitatively different in two models of quantum dynamics with time-dependent randomness: the Brownian SYK model [20, 21] (Section 4) and a quantum automaton circuit (Section 5). In these models, the value of $\gamma$ effectively doubles: $\gamma \rightarrow 2\gamma$. As we will carefully explain, the discrepancy between the Hamiltonian quantum dynamics and the random time-dependent quantum dynamics is that the former relies on many-body quantum coherence effects, while the latter does not. This slowdown in effectively classical operator growth processes relative to quantum-coherent operator growth processes is reminiscent of the quadratic speed up of quantum walks over classical random walks [22, 23].

## 2 Preliminaries

### 2.1 Hilbert space

In this paper, we study quantum many-body systems with Hilbert space

$$\mathcal{H} = (\mathbb{C}^2)^{\otimes N}. \tag{2.8}$$

We interpret each copy of $\mathbb{C}^2 = \text{span}(|0\rangle, |1\rangle)$ as consisting of either an empty site $|0\rangle$ or an occupied site $|1\rangle$. We define the density operators

$$n_i = |1_i\rangle\langle 1_i|, \tag{2.9}$$

which measure whether the site $i = 1, \ldots, N$ is occupied, together with the total conserved charge

$$Q = \sum_{i=1}^{N} n_i. \tag{2.10}$$

The Hilbert space $\mathcal{H}$ can be written as the direct sum of subspaces with a fixed number of up spins:

$$\mathcal{H} = \bigoplus_{N^\uparrow = 1}^{N} \mathcal{H}^{N^\uparrow}, \tag{2.11}$$

with

$$\mathcal{H}^{N^\uparrow} = \text{span}\left\{|n_1 n_2 \cdots n_N\rangle : \sum_{i=1}^{N} n_i = N^\uparrow\right\}. \tag{2.12}$$

Let $U(t)$ be a time-dependent unitary transformation on $\mathcal{H}$. In this paper, we are interested in studying the growth of operators when

$$[U(t), Q] = 0, \tag{2.13}$$

namely charge is conserved. The fact that charge is conserved means that the dynamics will separate the Hilbert space into $N + 1$ sectors corresponding to the allowed values of $Q = 0, 1, \ldots, N$. In this paper, we will be interested in quantum dynamics in subspaces when $Q$ and $N$ are taken to be very large, while the ratio

$$\bar{n} = \frac{Q}{N} \tag{2.14}$$

is held fixed. We will refer to $\bar{n}$ as the charge density, and focus on the limit $\bar{n} \ll 1$.

## 2.2 Operator dynamics and operator size

This paper is about operator growth: intuitively, given an operator such as $n_i$ which initially acts on only a finite number of sites (in this case 1), how much does time evolution scramble the information in $n_i$? Put another way, how complicated is the operator $n_i(t)$? To answer this question carefully, we introduce a new formalism, following [24]. Let $\mathcal{B}$ denote the space of operators acting on $\mathcal{H}$. For any operator $A \in \mathcal{B}$, time evolution is defined by

$$A(t) := U(t)^\dagger A U(t). \tag{2.15}$$

It is useful to write elements $A \in \mathcal{B}$ as "kets" $|A)$ to emphasize linearity of quantum mechanics on operators, which will play a critical role in this paper. When $t$ is a continuous parameter (i.e. we are not studying quantum circuit dynamics) we can also define the Liouvillian

$$\frac{\mathrm{d}}{\mathrm{d}t}|A(t)) := \mathcal{L}(t)|A(t)). \tag{2.16}$$

It is obvious that charge conservation places some constraints on operator growth. An operator which takes states of charge $Q$ to states of charge $Q'$ will do so at all times. However, at finite $\overline{n}$, there are $O(\exp(N))$ such operators, so this constraint is not immediately useful or physically illuminating.

The purpose of this paper is to present a better way of thinking about operator growth in such systems, at a given density $\overline{n}$. To do so, it is helpful to switch to a (grand) canonical perspective, and think about fixed chemical potential $\mu$ rather than fixed charge $Q$. Let

$$\rho = \frac{\mathrm{e}^{-\overline{\mu}Q}}{(1 + \mathrm{e}^{-\overline{\mu}})^N}, \qquad \text{where} \quad \overline{\mu} := \lim_{\beta \to 0} \beta\mu \tag{2.17}$$

denote the (grand) canonical density matrix at chemical potential $\mu$, normalized so that $\mathrm{tr}(\rho) = 1$. Here we introduce the dimensionless chemical potential $\overline{\mu}$, which is the physically meaningful quantity in the infinite temperature limit. Note that

$$\overline{n} = \frac{1}{1 + \mathrm{e}^{\overline{\mu}}}. \tag{2.18}$$

Given $\rho$, we now define the following inner product on $\mathcal{B}$:

$$(A|B) := \mathrm{tr}\left(\sqrt{\rho} A^\dagger \sqrt{\rho} B\right). \tag{2.19}$$

For any value of $\mu$, the length of an operator, which we define as $(A|A)$, does not grow:

$$(A(t)|A(t)) = (A|A) \tag{2.20}$$

since $[\rho, U] = 0$ following (2.13).

Equipped with this inner product, we are now ready to define a physically useful notion of operator size and operator growth at finite $\overline{\mu}$. There are two possible interpretations of $\mathcal{H}$, either in the language of spin models with a conserved $z$-spin, or in the language of fermion models with conserved charge. A non-local Jordan-Wigner-type transformation can convert between the two, but operator dynamics is *not* invariant under this transformation. For almost every quantum system [25], dynamics will only appear local in one language. So while there are clear similarities between how we talk about operator size and operator growth for a bosonic system vs. fermionic system, we must discuss each separately.

Let us first describe the physics when we interpret the Hilbert space in terms of bosonic degrees of freedom. First consider a single copy of $\mathbb{C}^2$ – i.e. a single site. There are 4 linearly

independent operators acting on this two level system, forming the span of the operator vector space $\mathcal{B}_i$. With respect to the inner product (2.19), an orthogonal set of them is

$$|1) = |0\rangle\langle 0| + |1\rangle\langle 1|, \tag{2.21a}$$

$$|X^+) = |1\rangle\langle 0|, \tag{2.21b}$$

$$|X^-) = |0\rangle\langle 1|, \tag{2.21c}$$

$$|n) = (1-\overline{n})|1\rangle\langle 1| - \overline{n}|0\rangle\langle 0|. \tag{2.21d}$$

The lengths of these operators are

$$(1|1) = 1, \tag{2.22a}$$

$$(X^+|X^+) = (X^-|X^-) = \frac{e^{-\overline{\mu}/2}}{1 + e^{-\overline{\mu}}}, \tag{2.22b}$$

$$(n|n) = \frac{e^{-\overline{\mu}}}{(1 + e^{-\overline{\mu}})^2}. \tag{2.22c}$$

For reasons that will become clear as we go through this paper, we define the size "superoperator" $\mathbb{S}$ as a linear transformation on $\mathcal{B}_i$:

$$\mathbb{S}|1) = 0|1), \tag{2.23a}$$

$$\mathbb{S}|X^+) = |X^+), \tag{2.23b}$$

$$\mathbb{S}|X^-) = |X^-), \tag{2.23c}$$

$$\mathbb{S}|n) = 2|n). \tag{2.23d}$$

Note also the useful identity

$$\frac{e^{-\overline{\mu}/2}}{1 + e^{-\overline{\mu}}} = \sqrt{\overline{n}(1-\overline{n})}. \tag{2.24}$$

If instead, the degrees of freedom are fermions, then on a single $\mathbb{C}^2$ the four orthogonal operators are $1, c, c^\dagger, c^\dagger c - \overline{n}$, where $c$ and $c^\dagger$ are usual creation and annihilation operators obeying

$$\{c, c^\dagger\} = 1. \tag{2.25}$$

The generalization of (2.22) holds. The definition of size is now slightly more intuitive, as it counts the number of creation and annihilation operators: now denoting $|n) = c^\dagger c - \overline{n}$:

$$\mathbb{S}|1) = 0|1), \tag{2.26a}$$

$$\mathbb{S}|c) = |c), \tag{2.26b}$$

$$\mathbb{S}|c^\dagger) = |c^\dagger), \tag{2.26c}$$

$$\mathbb{S}|n) = 2|n). \tag{2.26d}$$

Thus far we have defined the size superoperator acting on a single Hilbert space, but it is straightforward to extend it to the $N$-body Hilbert space. Letting $|T^a)$ ($a = 1, \dots, 4$) denote the four orthogonal operators above with length $L_a$ given in (2.22) and size $S_a$ given in (2.23) or (2.26) on a single two-level system, we observe that the following is an orthogonal basis for $\mathcal{B}$:

$$\mathcal{B} = \bigotimes_{i=1}^{N} \operatorname{span}\{T_i^a\} := \operatorname{span}\{| \otimes_i T_i^a)\}. \tag{2.27}$$

The length of each operator is

$$(\otimes_i T_i^a | \otimes_i T_i^a) = \prod_{i=1}^{N} L_{a,i}. \tag{2.28}$$

Size is then defined as

$$\mathbb{S}|\otimes_i T_i^a) = \left(\sum_{i=1}^{N} S_{a,i}\right)|\otimes_i T_i^a). \tag{2.29}$$

Let $\mathbb{Q}_s$ denote a projector onto many-body operators of size $s$. Due to (2.20), we may define the probability that the operator $A$ has size $s$ at time $t$ to be

$$P_s(t) := \frac{(A(t)|\mathbb{Q}_s|A(t))}{(A|A)}. \tag{2.30}$$

See [26] for a different interpretation of size at finite density or temperature.

The probability that an operator has size $s$ is related to the more convential out-of-time-ordered correlation functions (OTOCs) which have been used to probe many-body chaos. As a simple example, let us consider a fermionic system, and ask for the typical magnitude of the normalized OTOC

$$C_{ij}(t) = \frac{\mathrm{tr}\left(\sqrt{\rho}[c_i^\dagger c_i, c_j(t)]^\dagger \sqrt{\rho}[c_i^\dagger c_i, c_j(t)]\right)}{\mathrm{tr}\left(\sqrt{\rho} c_j \sqrt{\rho} c_j^\dagger\right)} \tag{2.31}$$

for different spins $i$. Since $[c_i^\dagger c_i, c_i] = -c_i$ and $[c_i^\dagger c_i, c_i^\dagger] = c_i^\dagger$, we conclude that this commutator will be non-vanishing whenever an operator string has either a $c_i$ or $c_i^\dagger$ on site $i$. Therefore,

$$\sum_{i=1}^{N} C_{ij}(t) \le \frac{(c_j(t)|\mathbb{S}|c_j(t))}{(c_j|c_j)}. \tag{2.32}$$

If the operator $c_j(t)$ did not have any strings with $c_i^\dagger c_i$ on any site, then (2.32) would be an equality. We conclude that just as in the uncharged models [24, 27], a typical OTOC $C_{ij}$ between a randomly chosen fermion $i$ and our initial fermion $j$ is non-vanishing only when the average operator size of $c_j(t)$ is large. However, crucially, this is when the operator size is measured with respect to the non-trivial inner product (2.19) at finite $\mu$.

# 3 Bounds on dynamics

In the limit $\overline{n} \ll 1$, which corresponds to $\overline{\mu} \gg 1$, we can estimate the canonical operators of size $s$ (in our basis) as having length $\sim \overline{n}^{s/2}$. Recall that the "length" here refers to the Frobenius-like norm of the operator in the finite $\overline{\mu}$ ensemble (2.19), whereas size counts the number of non-identity operators (with the inner product described above). As we now show, the fact that the canonical operators of size $s$ have an exponentially small length leads to a significant slowdown in the dynamics of our operator size.

## 3.1 Lyapunov exponent

For illustrative purposes, we focus on Hamiltonian quantum dynamics generated by the fermionic $q$-body (also called $q$-local) Hamiltonian (note $q$ must be even)

$$H(t) = \mathrm{i}^{\frac{q}{2}} \sum_{i_1 < \ldots < i_{q/2}, j_1 < \ldots < j_{q/2},} J_{i_1 \cdots i_{q/2} j_1 \cdots j_{q/2}} c_{i_1}^\dagger \cdots c_{i_{q/2}}^\dagger c_{j_1} \cdots c_{j_{q/2}}. \tag{3.33}$$

If the $J$ are all random, and are appropriately normalized, then this model is the complex SYK model [18,19,28] described in the next section. But we can also consider a more general class of models.

Now let us start with the operator $c_j$, as in our previous discussion. Our goal is to bound $P_s(t)$. In general, this is a challenging task [24], and requires finding the maximal eigenvalue of $\mathbb{Q}_s \mathcal{L} \mathbb{Q}_{s'}$. For illustrative purposes, it suffices to focus on what happens when $s' = 1$ and $s = q - 1$. Without loss of generality,[1] consider the size 1 operator

$$\mathcal{O}_1 = \sum_{i=1}^{N} a_i c_i, \tag{3.34}$$

normalized so that

$$\sum_k |a_k|^2 = 1. \tag{3.35}$$

Observe that

$$\frac{(\mathcal{O}_1 | \mathcal{L}^\mathsf{T} \mathbb{Q}_{q-1} \mathcal{L} | \mathcal{O}_1)}{(\mathcal{O}_1 | \mathcal{O}_1)} \leq \sum_{\substack{i_1, \dots i_{q/2-1} \\ j_1, \dots, j_{q/2}}} \left| \sum_k J_{k i_1 \cdots i_{q/2-1} j_1 \cdots j_{q/2}} a_k \right|^2 \frac{(c_{i_1}^\dagger \cdots c_{i_{q/2-1}}^\dagger c_{j_1} \cdots c_{j_{q/2}} | c_{i_1}^\dagger \cdots c_{i_{q/2-1}}^\dagger c_{j_1} \cdots c_{j_{q/2}})}{(c_k | c_k)}$$

$$= \frac{1}{(2 \cosh \frac{\overline{\mu}}{2})^{q-2}} \sum_{\substack{i_1, \dots i_{q/2-1} \\ j_1, \dots, j_{q/2}}} \left| \sum_k J_{k i_1 \cdots i_{q/2-1} j_1 \cdots j_{q/2}} a_k \right|^2. \tag{3.36}$$

Now, the summation above does not depend on $\mu$. The maximal eigenvalue of $\mathbb{Q}_{q-1} \mathcal{L} \mathbb{Q}_1$ corresponds to maximizing the sum, which can be done independently of $\mu$. Moreover, this argument did not depend on the choice of sizes $s$ and $s'$. We conclude that the maximal eigenvalue of $\mathbb{Q}_s \mathcal{L} \mathbb{Q}_{s'}$, denoted as $\|\mathbb{Q}_s \mathcal{L} \mathbb{Q}_{s'}\|$, obeys

$$\|\mathbb{Q}_s \mathcal{L} \mathbb{Q}_{s'}\| \leq \frac{\|\mathbb{Q}_s \mathcal{L} \mathbb{Q}_{s'}\|_{\overline{\mu}=0}}{\sqrt{\cosh^{|s-s'|} \frac{\overline{\mu}}{2}}}. \tag{3.37}$$

Note the square root above, which arises due to the fact that there were two $\mathcal{L}$s in (3.36). Therefore, the growth of larger operators from smaller operators is parametrically slowed down at large $|\overline{\mu}|$, or when the system becomes low or high density.

Note that (3.37) holds whether or not $s < s'$ or $s > s'$. After all, for a charge conserving system, $H$ commutes with $\rho$, and so $(A|\mathcal{L}|B) = -(B|\mathcal{L}|A)$. $\mathbb{Q}_s \mathcal{L} \mathbb{Q}_{s'}$ and $\mathbb{Q}_{s'} \mathcal{L} \mathbb{Q}_s$ are transposes, and have the same maximal singular value (i.e. operator norm).

A quick route to justifying (3.37) is to simply observe from (2.22) that operators of size $r$ always have their length reduced by a factor of $(\operatorname{sech} \frac{\overline{\mu}}{2})^r$ compared to what we might have naively expected based on the conventional Frobenius ($\overline{\mu} = 0$) inner product. Still, the reason that (3.37) is not trivial is that as we change the value of $\overline{\mu}$, the definition of $|n)$ also changes, and so which operators have a given size must also change! After all, we know that the probability distribution $P_s(t)$ – which does have the $\overline{\mu}$-rescaled lengths built into it – is a well-defined probability distribution at any $\overline{\mu}$, and this would simply be impossible if our procedure was nothing more than re-scaling the lengths of strings of $r$ $c$ and $c^\dagger$ operators by an $r$-dependent factor. The remarkable feature of all charge-conserving dynamics is that the operator evolution proceeds in just the right way so as to ensure the cancellation of two $\overline{\mu}$-dependent changes to our prescription: the change in the size-2 operator $|n)$, and the change in the inner product (2.19).

Having now understood the physics behind (3.37), we can now immediately apply it to problems of interest. We begin by discussing the Lyapunov exponent of infinite temperature

---

[1]We can ignore $c_i^\dagger$ contributions as they will always be orthogonal under time evolution, as operators which change $Q$ by different amounts are always orthogonal.

fermionic theories of the form (3.33) with a U(1) conservation law, defined by the exponential growth of (2.31):

$$C_{ij}(t) \sim \frac{1}{N} e^{\lambda_L t} \tag{3.38}$$

for times smaller than the scrambling time $t \sim \lambda_L^{-1} \log N$. As we described in (2.32), the growth of $C_{ij}(t)$ for generic $i$ and $j$ arises from the growth in effective $\mu$-dependent size of the operator. Since the growth rates in size have been rescaled by (3.37), and due to charge conservation operator size can only grow by an even number, we can immediately find the *universal* bound

$$\lambda_L \leq \sqrt{4\overline{n}(1-\overline{n})}\lambda_*. \tag{3.39}$$

Here $\lambda_*$ is a constant which comes from the $\overline{\mu} = 0$ bounds, following [24]. The $\overline{n}$ scaling above comes from applying (2.24) to (3.37). The point of this paper is not the evaluation of $\lambda_*$, which can be quite challenging, but rather in the universal $\overline{n}$ dependence of (3.39).

A key ingredient in (3.39) is that operator size can only grow by an even number. In the fermion language, for example, we understand this as follows: the size 1 operators are $c_i$ and $c_i^\dagger$. Suppose we have an operator $A$ which transitions between the Hilbert spaces $\mathcal{H}^{Q_1}$ and $\mathcal{H}^{Q_2}$ at fixed charges $Q_1$ and $Q_2$, as defined in (2.11). Each time we multiply by a product of creation/annihilation operators of size $s$, $|Q_1 - Q_2|$ (mod 2) changes by an amount $s$ (mod 2). Hence, $A(t)$ will only involve operators whose size is either all even or all odd. This result also holds in the spin language.

In certain models with all-to-all interactions, including the SYK model, we can parametrically improve upon (3.39). In the SYK model, operator growth in the large $N$ limit is dominated by processes that grow operators by $q - 2$ $c$ and $c^\dagger$ at a time [24,27]. In this case, we can strengthen (3.39) to

$$\lambda_L(\overline{\mu}) \leq (4\overline{n}(1-\overline{n}))^{(q-2)/4} \lambda_*. \tag{3.40}$$

It is interesting that our approach readily leads to density-dependent bounds on Lyapunov exponents, which appear challenging to derive by other means [29]. We also emphasize that (3.40) does not depend on the precise choice of operators used in the OTOC.

## 3.2 Butterfly velocity

A more conjectural application of our rigorous result (3.37) is to constrain a suitably defined butterfly velocity. Consider a $d$-dimensional fermionic many-body system on a lattice of the form

$$H = \sum_{\text{local sets} X_1, X_2} J_X \prod_{i \in X_1} c_i^\dagger \prod_{i \in X_2} c_i, \tag{3.41}$$

where the sum $X$ runs over sufficiently local sets (e.g. no two sites in $X_{1,2}$ are farther than $m$ sites apart, where $m$ is some O(1) number). Charge conservation means that $|X_1| = |X_2|$ in the sum above. Let us define $v_B^*$ as follows: in a chaotic system, an operator $c_j(t)$ grows such that a typical OTOC $C_{ij}(t)$ is order 1 inside a ball of radius $v_B^* t$ around site $j$. We propose for a generic system that there exists a constant $v_B^*$ such that

$$v_B(\mu) \leq (4\overline{n}(1-\overline{n}))^{(q-2)/4} v_B^*. \tag{3.42}$$

The exponent $\frac{q-2}{4}$ above should be understood in the same context as (3.40): in the worst case scenario, we should set this exponent to $\frac{1}{2}$, however in certain models it may be possible to improve the exponent to $\frac{q-2}{4}$.

To (heuristically) obtain (3.42), we use a technique from [30]. For simplicity, assume that we have an operator supported at the origin of a $d$-dimensional lattice, $\mathcal{O}_0$, and that all terms in the Hamiltonian are either single-site fields, or nearest-neighbor interactions. We are

interested in the weight of this operator on a site $j$ a distance $d_j$ from the origin. So, let us define the superoperator

$$\mathcal{F} = \sum_j e^{d_j} \mathbb{S}_j, \tag{3.43}$$

where $\mathbb{S}_j$ denotes the size of an operator on lattice site $j$. We now bound

$$\frac{d}{dt}(\mathcal{O}_0(t)|\mathcal{F}|\mathcal{O}_0(t)) = (\mathcal{O}_0(t)|[\mathcal{F},\mathcal{L}]|\mathcal{O}_0(t)) \le \sum_{R,R'}(\mathcal{F}_R - \mathcal{F}_{R'})(\mathcal{O}_0(t)|\mathbb{Q}_R \mathcal{L} \mathbb{Q}_{R'}|\mathcal{O}_0(t)), \tag{3.44}$$

where $\mathbb{Q}_R$ denotes a projection onto operators which have support on site $i$ if and only if $i \in R$. Then,

$$\frac{d}{dt}(\mathcal{O}_0(t)|\mathcal{F}|\mathcal{O}_0(t)) \lesssim \sum_{R,R'} \frac{|\mathcal{F}_R - \mathcal{F}_{R'}|}{2} \frac{\|\mathbb{Q}_R \mathcal{L} \mathbb{Q}_{R'}\|_{\overline{\mu}=0}}{\cosh^{(q-2)/2} \frac{\overline{\mu}}{2}} ((\mathcal{O}_0(t)|\mathbb{Q}_R|\mathcal{O}_0(t)) + (\mathcal{O}_0(t)|\mathbb{Q}_{R'}|\mathcal{O}_0(t))), \tag{3.45}$$

where we have used the Cauchy-Schwarz inequality and used the $\mu$-dependent inner product similarly to (3.37). However, this step is *not* rigorous because we will not prove that nearly all weight corresponds to sets $R$ and $R'$ that differ in $q-2$ sites (or even just $q-2$ fermions in the operator). Nevertheless proceeding with our argument, the key observation is that spatial locality on the lattice demands there exists a finite positive constant $K$ such that for any two sets $R$ and $R'$ contained in (3.44),

$$|\mathcal{F}_R - \mathcal{F}_{R'}| \le K \min_{i \in R \cap R'} e^{d_i}. \tag{3.46}$$

Moreover, the sum over sets $R$ and $R'$ which share a union $R \cap R' = i$ is finite. Therefore, we may replace the sum over $R$ and $R'$ by a sum over lattice sites $i$:

$$\frac{d}{dt}(\mathcal{O}_0(t)|\mathcal{F}|\mathcal{O}_0(t)) \lesssim \sum_i 2zK e^{d_i} \frac{\|\mathbb{Q}_R \mathcal{L} \mathbb{Q}_{R'}\|_{\overline{\mu}=0}}{\cosh^{(q-2)/2} \frac{\overline{\mu}}{2}} ((\mathcal{O}_0(t)|\mathbb{Q}_i|\mathcal{O}_0(t)))$$
$$\lesssim \frac{K'}{\cosh^{(q-2)/2} \frac{\overline{\mu}}{2}} \sum_i e^{d_i}(\mathcal{O}_0(t)|\mathbb{Q}_i|\mathcal{O}_0(t)) = \frac{K'}{\cosh^{(q-2)/2} \frac{\overline{\mu}}{2}}(\mathcal{O}_0(t)|\mathcal{F}|\mathcal{O}_0(t)), \tag{3.47}$$

where $z$ is another O(1) constant related to the number of sets $X_{1,2}$ in $H$ containing site $i$, and $K'$ is yet another O(1) constant. Hence we conclude that

$$(\mathcal{O}_0(t)|\mathcal{F}|\mathcal{O}_0(t)) \lesssim \exp\left[\frac{K't}{\cosh^{(q-2)/2} \frac{\overline{\mu}}{2}}\right]. \tag{3.48}$$

However, comparing with (3.43), and using Markov's inequality [30], we conclude that

$$(\mathcal{O}_0(t)|\mathbb{Q}_x|\mathcal{O}_0(t)) \lesssim \exp\left[\frac{K't}{\cosh^{(q-2)/2} \frac{\overline{\mu}}{2}} - d_x\right], \tag{3.49}$$

which implies (3.42).

It is straightforward to generalize these results to spin models, rather than fermionic models. In models that are not of the form (3.33) and involve couplings with multiple different "$q$" (i.e. numbers of fermions), then the $q$ in the above bounds should be replaced with the smallest value of $q > 2$ that appears in the Hamiltonian (since $q = 2$ terms do not grow operators).

# 4 Charged SYK model and its Brownian version

In this section, we consider two versions of the SYK model: one with Brownian motion couplings [20, 21], and one with time-independent couplings. We will see that the behavior of operator growth in these two models qualitatively differs when $\overline{n} \ll 1$.

## 4.1 General methodology

The Brownian SYK and the regular SYK have the same form of the Hamiltonian (3.33) with a different random ensemble for the coupling constants (we upgrade the coupling $J \to J(t)$ to be generally time dependent):

$$\text{regular} \quad \langle J_{j_1 \dots j_q}(t) J^*_{j'_1 \dots j'_q}(t') \rangle = \delta_{j_1 j'_1} \dots \delta_{j_q j'_q} \frac{J^2 (q/2-1)!(q/2)!}{N^{q-1}}, \tag{4.50a}$$

$$\text{Brownian} \quad \langle J_{j_1 \dots j_q}(t) J^*_{j'_1 \dots j'_q}(t') \rangle = \delta_{j_1 j'_1} \dots \delta_{j_q j'_q} \delta(t-t') \frac{J(q/2-1)!(q/2)!}{N^{q-1}}. \tag{4.50b}$$

Note that in both cases above, $J$ has the units of energy.

We now wish to calculate the Lyapunov exponent at fixed $\overline{\mu} = \beta \mu$, as we take the infinite temperature limit $\beta \to 0$. Unfortunately, neither of the analytically controlled limits of the SYK model – the strong coupling limit $T \ll J$, or the large $q$ limit – can be directly applied for our problem. After all, we are interested in $\beta = 0$ in this paper, invalidating the former approach. Moreover, if $q \to \infty$, we can expect from (3.40) that for $\overline{n} < \frac{1}{2}$, the calculation becomes trivial: $\lambda_L$ will vanish at leading order since for any $0 < c < 1$, $c^q \to 0$ as $q \to \infty$. This means the latter approach also is not directly useful.

Ultimately, we rely on an approximate method, which we expect will miss O(1) factors for the time-independent SYK model, but will otherwise be accurate. For the Brownian SYK model, however, our results will be exact. We use the Keldysh formalism with the assumption of a quasi-particle-like Green's function

$$G^R(\omega) \approx \frac{1}{\omega + \mu + i\Gamma}, \tag{4.51}$$

where $\Gamma$ is the quasi-particle decay rate that will be self-consistently estimated. With the above approximated form of retarded Green function, we will find the Lyapunov exponent via the following kinetic equation [19, 31, 32]:

$$G^R\left(\omega + i\frac{\lambda_L}{2}\right) G^A\left(\omega - i\frac{\lambda_L}{2}\right) \int \frac{d\omega'}{2\pi} R(\omega - \omega') F^R(\omega') = F^R(\omega), \tag{4.52}$$

where $F^R$ stands for the vertex function that contains the information of OTOC as a function of relative time (not the center of mass time which has been characterized by the $\lambda_L$ here). $R(\omega)$ is the rung function $R = \delta \Sigma^K / \delta G^K$ obtained in the Keldysh formalism via the input $G^R$ we have in (4.51). We adopt a commonly used approximation [31]

$$G^R\left(\omega + i\frac{\lambda_L}{2}\right) G^A\left(\omega - i\frac{\lambda_L}{2}\right) \approx 2\pi \delta(\omega + \mu) \cdot \frac{1}{\lambda_L + 2\Gamma}. \tag{4.53}$$

Therefore $F^R(\omega) \approx \delta(\omega + \mu)$, and we have obtained

$$\lambda_L + 2\Gamma = R(0) \tag{4.54}$$

here $R(0)$ is the zero frequency component of the rung function.

Before applying the above formulas to the regular and Brownian SYK, let us clarify the validity of the approach used here. As is commented in Ref. [33], the above procedure is an approximation method for regular SYK because (*1*) in general the determination of the self energy at IR requires full knowledge of the Green function, not just the IR and UV limits. Therefore, the prefactor of the quasi-particle decay rate $\Gamma$ is not expected to be accurate, while the scaling is still expected to be valid. (*2*) the approximation (4.53) also introduces inaccuracy for the prefactor of $\Gamma$. However, the above two sources of error will not occur for the Brownian SYK, because (*1*) the interaction is localized in time, so the self energy can be determined by the UV of the Green's function completely; (*2*) the relation (4.54) can be justified when $R(\omega) = R(0)$ is a constant in frequency. One way to see this is to rewrite (4.52) as follows

$$\frac{1}{(\omega+\mu)^2 + \left(\Gamma + \frac{\lambda_{\mathrm{L}}}{2}\right)^2} R(0) \int \frac{\mathrm{d}\omega'}{2\pi} F^{\mathrm{R}}(\omega') = F^{\mathrm{R}}(\omega). \tag{4.55}$$

Integrating over $\omega$ for both sides, and eliminating the integral, we obtain (4.54).

## 4.2 Time-independent (regular) SYK

We first study the time-independent (regular) SYK model. The first step is to obtain $\Gamma$ self-consistently. By definition, we have $\mathrm{i}\Gamma = -\Sigma^{\mathrm{R}}(\omega \to -\mu)$, and the retarded self energy can be obtained via Schwinger-Dyson equations (see Appendix A) and the the assumed form of $G^{\mathrm{R}}$. In the limit $\beta \to 0$ at fixed $\beta\mu = \overline{\mu}$ and $\Gamma$, we have

$$\Sigma^{\mathrm{R}}(t) \approx -\mathrm{i}\Theta(t) \frac{J^2}{(2\cosh\frac{\overline{\mu}}{2})^{q-2}} \mathrm{e}^{-(q-1)\Gamma t} \mathrm{e}^{\mathrm{i}\mu t}, \tag{4.56}$$

whose $\omega \to -\mu$ component is

$$\Sigma^{\mathrm{R}}(\omega \to -\mu) = \frac{J^2}{(2\cosh\frac{\overline{\mu}}{2})^{q-2}} \frac{-\mathrm{i}}{(q-1)\Gamma}. \tag{4.57}$$

Now, equating the above expression with $-\mathrm{i}\Gamma$, we obtain

$$\Gamma \approx \frac{J}{\sqrt{q-1}(2\cosh\frac{\overline{\mu}}{2})^{\frac{q-2}{2}}}. \tag{4.58}$$

However, as we commented above, we should not trust the constant prefactor in $\Gamma$ for general $q$.[2] What is important is the dependence on $J$ and $\overline{\mu}$, therefore for the rest of this subsection we will drop the unimportant prefactors.

Next, we compute the rung function

$$R(t) = \frac{\delta \Sigma^{\mathrm{K}}}{\delta G^{\mathrm{K}}} = (q-1)\frac{J^2}{2^{q-2}} \left(G^{\mathrm{K}}_{21}(t)G^{\mathrm{K}}_{12}(-t)\right)^{\frac{q-2}{2}} \sim J^2 \frac{\mathrm{e}^{-(q-2)\Gamma|t|}}{(2\cosh\frac{\overline{\mu}}{2})^{q-2}}. \tag{4.59}$$

Thus, the $J$ and $\overline{\mu}$ dependence for its zero frequency component is given as follows

$$R(\omega \to 0) \sim \frac{J^2}{\Gamma(2\cosh\frac{\overline{\mu}}{2})^{q-2}} \sim \frac{J}{(2\cosh\frac{\overline{\mu}}{2})^{\frac{q-2}{2}}}. \tag{4.60}$$

Recall that within our approximation method, the Lyapunov exponent (4.54) is a linear combination of $R(0)$ and $\Gamma$, so we conclude that

$$\lambda_{\mathrm{L}} \approx R(0) - 2\Gamma \sim \frac{J}{(2\cosh\frac{\overline{\mu}}{2})^{\frac{q-2}{2}}}. \tag{4.61}$$

---

[2]The prefactor is expected to be accurate only at $q \to 2$.

In terms of charge filling $\overline{n}$, we have

$$\lambda_{\mathrm{L}}(\overline{n}) = (4\overline{n}(1-\overline{n}))^{(q-2)/4} \lambda_{\mathrm{L}}\left(\frac{1}{2}\right), \tag{4.62}$$

which saturates our general bound (3.40).

## 4.3 Brownian SYK

Next, we will move to the Brownian SYK where we will see a different scaling w.r.t $\cosh\frac{\overline{\mu}}{2}$. The computational logic for Brownian SYK is the same as for the regular SYK model; the only difference is that the interaction is uncorrelated in time. As a consequence, the two approximations in the above section become exact. For example, the self energy

$$\Sigma^{\mathrm{R}}(t) = -\mathrm{i}\Theta(t)\frac{J\delta(t)}{(2\cosh\frac{\overline{\mu}}{2})^{q-2}}\mathrm{e}^{-(q-1)\Gamma t}\mathrm{e}^{\mathrm{i}\mu t} = -\mathrm{i}\Theta(t)\frac{J\delta(t)}{(2\cosh\frac{\overline{\mu}}{2})^{q-2}} \tag{4.63}$$

only relies on the UV behavior of the Green's function. Its Fourier transform[3] is a constant:

$$\Sigma^{\mathrm{R}}(\omega) = -\mathrm{i}\frac{J}{2^{q-1}\cosh^{q-2}\frac{\overline{\mu}}{2}}. \tag{4.64}$$

Thus,

$$\Gamma := \mathrm{i}\Sigma^{\mathrm{R}}(-\mu) = \frac{J}{2^{q-1}\cosh^{q-2}\frac{\overline{\mu}}{2}}. \tag{4.65}$$

Comparing with (4.58), we notice that the power law exponent of $\cosh\frac{\overline{\mu}}{2}$ is twice that of the regular SYK model.

Similarly, the rung function

$$R(t) = (q-1)\frac{J\delta(t)}{2^{q-2}}\left(G_{21}^{\mathrm{K}}(t)G_{12}^{\mathrm{K}}(-t)\right)^{\frac{q-2}{2}} = (q-1)J\delta(t)\frac{1}{(2\cosh\frac{\overline{\mu}}{2})^{q-2}}, \tag{4.66}$$

$$R(\omega \to 0) = \frac{(q-1)J}{(2\cosh\frac{\overline{\mu}}{2})^{q-2}}. \tag{4.67}$$

The Lyapunov exponent $\lambda_{\mathrm{L}} = R(0) - 2\Gamma$ is therefore obtained as follows

$$\lambda_{\mathrm{L}} = (q-2)\frac{J}{2^{q-2}}\frac{1}{(2\cosh\frac{\overline{\mu}}{2})^{q-2}} \propto (\overline{n}(1-\overline{n}))^{(q-2)/2}. \tag{4.68}$$

As commented before, this formula for the Brownian SYK is exact[4], and we also note that the power is twice the result in the regular SYK.

## 4.4 Physical comparison between regular SYK and Brownian SYK

Let us now give a few physical arguments for the discrepancy between the Brownian/regular SYK models, as we believe this physics is somewhat universal (especially in models related to holographic gravity).

---

[3]Note the expression involves a discontinuous function $\Theta(t)$ multiplying a delta function $\delta(t)$, and we need to take the average of $\Theta(t)$ from two sides.

[4]At $\mu = 0$, the result is consistent with Ref. [21] where the Lyapunov exponent is obtained using a completely different method.

In the regular SYK model, we can loosely think of the density-dependence of $\lambda_L$ as follows. Consider a Taylor expansion of a time evolving operator, which looks schematically like

$$c_1^\dagger(t) = c_1^\dagger + it[H, c_1^\dagger] + \cdots \sim c_1^\dagger + it \sum_{j_2,\ldots,j_q} J_{1,j_2,\ldots,j_q} c_{j_2} \cdots c_{j_{\frac{q}{2}}} c_{j_{\frac{q}{2}+1}}^\dagger \cdots c_{j_q}^\dagger + \cdots. \qquad (4.69)$$

In the first term of the Taylor series above, the operator has increased in size by $q-2$ $c$ and $c^\dagger$. By the formalism we developed above in (2.22), we know that each additional $c$ and $c^\dagger$ leads to an effective change in length of order $\bar{n}^{1/4}$. Recognizing that each subsequent commutator with $H$ adds $q-2$ more fermions, we can immediately see that the coefficient of $c_1(t)$ at order $t^k$ has length $\bar{n}^{k(q-2)/4}$, which immediately implies (3.40).

Alternatively, if we are at low density $\bar{n}$, then we can ask how many states there are which have a fermion on sites $j_2,\ldots,j_{\frac{q}{2}}$ – the second term in (4.69) will annihilate any state where even one of those sites is unoccupied. At low density, the fraction of such states is $\bar{n}$ per site. So we might estimate the disorder-averaged average size to be

$$\langle (c_1(t)|\mathbb{S}|c_1(t)) \rangle \approx \bar{n} \left( 1 + t^2 \sum_{j_2,\ldots,j_q} |J_{1,j_2\cdots j_q}|^2 \times (q-1)\bar{n}^{(q-2)/2} + \cdots \right). \qquad (4.70)$$

Again, the series above will be a function of $t\bar{n}^{(q-2)/4}$.

In the Brownian SYK, due to the time-dependent disorder average in (4.50b), we would instead find

$$\langle (c_1(t)|\mathbb{S}|c_1(t)) \rangle \approx \bar{n} \left( 1 + t \sum_{j_2,\ldots,j_q} |J_{1,j_2\cdots j_q}|^2 \times (q-1)\bar{n}^{(q-2)/2} + \cdots \right). \qquad (4.71)$$

Here, the series is a function of $t\bar{n}^{(q-2)/2}$, which heuristically explains the doubling of the Lyapunov exponent.

Ultimately, therefore, the difference between the Lyapunov exponents of the regular SYK model and the Brownian SYK model is the role of quantum coherence effects. Randomness in time, and not among the different coupling constants $J$, was responsible for the decoherence in the Brownian operator growth. This is analogous to the quadratic speed-up of coherent quantum walks over incohererent quantum walks, the latter of which behave identically to classical random walks [22,23]. Our universal bound (3.40) will be saturated by models, like SYK, with highly quantum coherent dynamics. It cannot be parametrically improved.

## 4.5 Butterfly velocity

We can generalize the discussions above to a spatially local version of the SYK model as introduced in [34,35]. Consider the Hamiltonian

$$H = \sum_{x,y} S_{xy} i^{\frac{q}{2}} \sum_{i_1 < \ldots < i_{q/2}, j_1 < \ldots < j_{q/2},} J_{i_1 \cdots i_{q/2} j_1 \cdots j_{q/2}}^{xy} c_{x,i_1}^\dagger \cdots c_{x,i_{q/2}}^\dagger c_{y,j_1} \cdots c_{y,j_{q/2}}, \qquad (4.72)$$

where the hopping matrix $S_{xy} \neq 0$ only if $x$ and $y$ are nearest neighbors, or $x = y$. For example, in one dimension, we could take

$$S_{xy} = \begin{cases} 1 - 2b & x = y \\ b & |x-y| = 1 \\ 0 & \text{otherwise} \end{cases}. \qquad (4.73)$$

The coefficients $J$ in (4.72) are defined so that $H$ is Hermitian. On this simple one dimensional lattice, the eigenvectors of $S_{xy}$ are plane waves $e^{ipx}$, with eigenvalues

$$S(p) = 1 - 2b(1 - \cos p). \tag{4.74}$$

The growth of OTOCs in space can be characterized by the hopping matrix $S_{xy}$ above, which enters the kinetic equation (4.52) in the following way

$$G^{R}\left(\omega + i\frac{\lambda_{L}}{2}\right)G^{A}\left(\omega - i\frac{\lambda_{L}}{2}\right)\int \frac{d\omega'}{2\pi}\sum_{y}S_{xy}R(\omega - \omega')F_{y}^{R}(\omega') = F_{x}^{R}(\omega). \tag{4.75}$$

Note that the spatial and temporal dependence are factorized. Therefore, we can directly diagonalize the hopping $S$ matrix using plane waves on the lattice. Within the approximation scheme we used before, we have the following $p$-dependent Lyapunov exponent

$$\lambda_{L}(p) + 2\Gamma = (1 - bp^{2})R(0) \Rightarrow \lambda_{L}(p) = \lambda_{L}(0) - bR(0)p^{2}, \tag{4.76}$$

where $\lambda_{L}(0) := \lambda_{L}(p \to 0)$ denotes the Lyapunov exponent we obtained in the case without spatial structure, while we remind that $R(0) := R(\omega \to 0)$ is the zero frequency component not the momentum.

In the weak coupling, the butterfly velocity is determined by the saddle point of the following Fourier transform[5]

$$F_{x}^{R}(t) \sim \int \frac{dp}{2\pi}e^{\lambda_{L}(0)t - bR(0)p^{2}t + ipx} \sim e^{\lambda_{L}(0)t - x^{2}/4bR(0)t} \tag{4.77}$$

from which we read out

$$v_{B}^{2} = 4b\lambda_{L}(0)R(0). \tag{4.78}$$

Regarding the dependence on the chemical potential/charge filling, we note that $R(0)$ and $\lambda_{L}(0)$ have the same dependence as we demonstrated in previous sections, therefore we conclude that $v_{B}$ scales in the same way as $\lambda_{L}$, namely

$$\frac{v_{B}(\overline{n})}{v_{B}(\frac{1}{2})} = \frac{\lambda_{L}(\overline{n})}{\lambda_{L}(\frac{1}{2})}. \tag{4.79}$$

This relation applies both to the regular and Brownian SYK. In particular, for the regular SYK, the above formula saturates the bound (3.42).

It is easy to show that the discussion above for the nearest neighbor one dimensional lattice – in particular, the conclusion (4.78), generalizes to any other lattice.

## 5 Random automaton circuit

In this section, we discuss a random quantum automaton (QA) circuit, composed of $N$ number of qubits (spin-$\frac{1}{2}$ degrees of freedom) with a global U(1) symmetry. Under QA dynamics, states expressed in the number basis (e.g. eigenstates of all Pauli $Z$ operators) are sent to other eigenstates, without generating quantum superposition. Due to this special property, QA circuits can be simulated using the classical Monte Carlo algorithm. They have been extensively used to study quantum dynamics in both integrable and chaotic systems with local interaction [37–41].

---

[5]For strong coupling $T \ll J$, there will be additional contributions to this integral [36].

## 5.1 Lyapunov exponent

Here, we construct a QA model consisting of $k$-qubit gates which acts on $k$ qubits randomly selected in the system. This model has all-to-all interactions, and at each time step, we apply roughly $N/k$ gates, to ensure extensive scaling of the dynamics in the large $N$ limit. We expect that under time evolution, this QA model exhibits similar operator growth to a large class of other random circuit models with U(1) symmetry, including Haar random circuits without locality [42,43] and the Brownian SYK model above.

In the QA circuit, the $k$-qubit gate is randomly chosen to be $U_k$ with probability $f$ or the identity with probability $1-f$. The $U_k$ gate is defined in the following way: For the $k$ number of qubits, if the middle one has $|1\rangle$ with the rest $k-1$ qubits having total $\langle Z \rangle = 0$, $U_k$ will flip all these $k-1$ qubits. It will leave other configurations invariant. The simplest case is $k=3$, where we have

$$U_3 \equiv 1 - |011\rangle\langle 011| - |110\rangle\langle 110| + |011\rangle\langle 110| + |110\rangle\langle 011|. \tag{5.80}$$

Clearly, this circuit conserves total $z$-spin, and is U(1)-symmetric. Similarly, if $k=5$, our QA circuit swaps between $|00111\rangle$ and $|11100\rangle$, $|10110\rangle$ and $|01101\rangle$, $|01110\rangle$ and $|10101\rangle$ and leaves other states invariant.

To understand the operator dynamics, we define the following operator basis for a single site:

$$P^\uparrow = |1\rangle\langle 1|, \tag{5.81a}$$

$$P^\downarrow = |0\rangle\langle 0|, \tag{5.81b}$$

$$X^+ = |1\rangle\langle 0|, \tag{5.81c}$$

$$X^- = |0\rangle\langle 1|. \tag{5.81d}$$

The space of many-body operators is a tensor product of this local basis. Any operator can be written as a superposition of these basis operators (Pauli string operators).

This choice follows [41] and differs from the choice made in (2.21); however, due to the non-Hamiltonian nature of the QA circuit, this choice will prove a little more convenient here. Under the $U_k$ gate, a Pauli string operator maps to another Pauli string operator.

Let $\mathcal{P}^{N^\uparrow}$ denote a projector onto the Hilbert space $\mathcal{H}^{N^\uparrow}$ defined in (2.11). Consider the operator dynamics for $X_x^+(t)\mathcal{P}^{N^\uparrow}$ in the limit $\bar{n} = N^\uparrow/N \ll 1$. In the operator basis defined in ((5.81)), $X_x^+(t=0)\mathcal{P}^{N^\uparrow}$ can be written as the superposition of Pauli strings with $N^\uparrow$ $P^\uparrow$s and $N-N^\uparrow-1$ $P^\downarrow$s. Under time evolution, the sum of the number of $X^+$ and $P^\uparrow$ remains invariant, as does the number of $X^-$ and $P^\downarrow$ together, due to charge conservation. Furthermore, the number of $X^+$ is always larger than $X^-$ by one. Operator growth can be characterized by counting the number of $X^+$ in $X_x^+(t)$, which is 1 at $t=0$ and eventually saturates to a value of order $N^\uparrow$.

Let us first assume $k=3$ and define the number of $X^+$ as $s$. Under random dynamics governed by our QA circuit, at early time, the most important update rule for the growth of $X^+$ is $P^\uparrow X^+ P^\downarrow \to X^+ X^+ X^-$. Notice that the probability for $P^\uparrow$, $X^+$ and $P^\downarrow$ are proportional to $\bar{n}$, $s$ and $1-\bar{n}$ respectively. Therefore in the continuous limit, we expect that

$$\frac{\mathrm{d}s}{\mathrm{d}t} \sim \bar{n}s, \tag{5.82}$$

which implies $s \sim \exp(\bar{n}t)$. The Lyapunov exponent $\lambda$ is proportional to the ratio $\bar{n}$. We can quickly generalize the above argument to any $k$. Since the probability to find $(k-1)/2$ $P^\uparrow$s (which, at low density, is the limiting constraint) is proportional to $\bar{n}^{(k-1)/2}$, the Lyapunov

exponent obeys

$$\lambda \sim \overline{n}^{(k-1)/2}. \tag{5.83}$$

In order to compare (5.83) to our general bound (3.40), we observe that the Hamiltonian which would generate the $U_k$ gate (by applying it for a finite time) is schematically)

$$\begin{aligned}
H_k &\sim X_1 \cdots X_{\frac{k-1}{2}} P^\uparrow_{\frac{k+1}{2}} X_{\frac{k+3}{2}} \cdots X_k \\
&\sim (X^+ + X^-)_1 \cdots (X^+ + X^-)_{\frac{k-1}{2}} n_{\frac{k+1}{2}} (X^+ + X^-)_{\frac{k+3}{2}} \cdots (X^+ + X^-)_k \\
&\quad + \overline{n}(X^+ + X^-)_1 \cdots (X^+ + X^-)_{\frac{k-1}{2}} (X^+ + X^-)_{\frac{k+3}{2}} \cdots (X^+ + X^-)_k,
\end{aligned} \tag{5.84}$$

where in the second step, we have switched (temporarily) to the operator basis (2.21). The operator on the first line is size $k+1$, while the operator on the second line is size $k-1$ but with an extra prefactor of $\overline{n}$. Hence, (3.40) would predict $\lambda_L \sim \overline{n}^{(k-1)/4}$. However, as we have already seen in Section 4.4, models with all-to-all interactions and time-dependent random couplings are not coherent enough to saturate (3.40), and their Lyapunov exponents scale with twice the power of $\overline{n}$. Upon accounting for this extra factor of 2, we reproduce (5.83).

We confirm this result numerically by computing OTOCs in the random QA circuit. For numerical ease, we study the following OTOC:

$$\begin{aligned}
C_{\mathrm{XZ}}^{ij}(t) &= -\frac{\mathrm{tr}\left\{\mathcal{P}^{N_\uparrow}\left[X_i(t), Z_j\right]^2\right\}}{\mathrm{tr}\mathcal{P}^{N_\uparrow}} = \sum_{s,s'} \frac{\left|\langle s|[X_i, Z_j(-t)]|s'\rangle\right|^2}{\mathrm{tr}\mathcal{P}^{N_\uparrow}} \\
&= \sum_s \frac{\left|\langle s|Z_j(-t)|s^*\rangle - \langle s^*|Z_j(-t)|s\rangle\right|^2}{\mathrm{tr}\mathcal{P}^{N_\uparrow}},
\end{aligned} \tag{5.85}$$

where $|s^*\rangle = X_i|s\rangle$ flips a single spin/bit.

We numerically computed $C_{\mathrm{XZ}}(t)$ by averaging over the index $i$ and $j$ of $C_{\mathrm{XZ}}^{ij}(t)$. As shown in Fig. 1(a) for the case $k=3$, $C_{\mathrm{XZ}}(t)$ increases exponentially at early times. The Lyapunov exponent $\lambda$ is linearly proportional to $\overline{n}$ when $\overline{n} \ll 1$. In Fig. 1(b), we show that the Lyapunov exponents of the $k=3,5,7$ QA circuits are consistent with (5.83) for $\overline{n} \ll 1$.

We further computed the two point auto correlation function

$$C_{\mathrm{O}}^i(t) = \frac{\mathrm{tr}\left\{\mathcal{P}^{N_\uparrow} O_i(t) O_i\right\}}{\mathrm{tr}\mathcal{P}^{N_\uparrow}}. \tag{5.86}$$

In terms of operator dynamics, this can be understood as the probability for the overlap between $\mathcal{P}^{N_\uparrow} O_i(t)$ and $O_i$ under time evolution, which should decay exponentially under the operator growth. As in the SYK models, we expect this decay rate is proportional to $\lambda_L$. As shown in Fig. 1(c), we numerically computed the averaged $C_Z$, and observed that

$$C_Z(t) - \left[\frac{\mathrm{tr}\left\{\mathcal{P}^{N_\uparrow} Z\right\}}{\mathrm{tr}\mathcal{P}^{N_\uparrow}}\right]^2 = C_Z(t) - (1 - 2\overline{n})^2 \sim \exp(-\kappa t). \tag{5.87}$$

Note that $(1 - 2\overline{n})^2$ is the saturation value $C_Z(\infty)$. As shown in Fig. 1(d), we find that

$$\kappa \sim \lambda_L \sim \overline{n}^{(k-1)/2}. \tag{5.88}$$

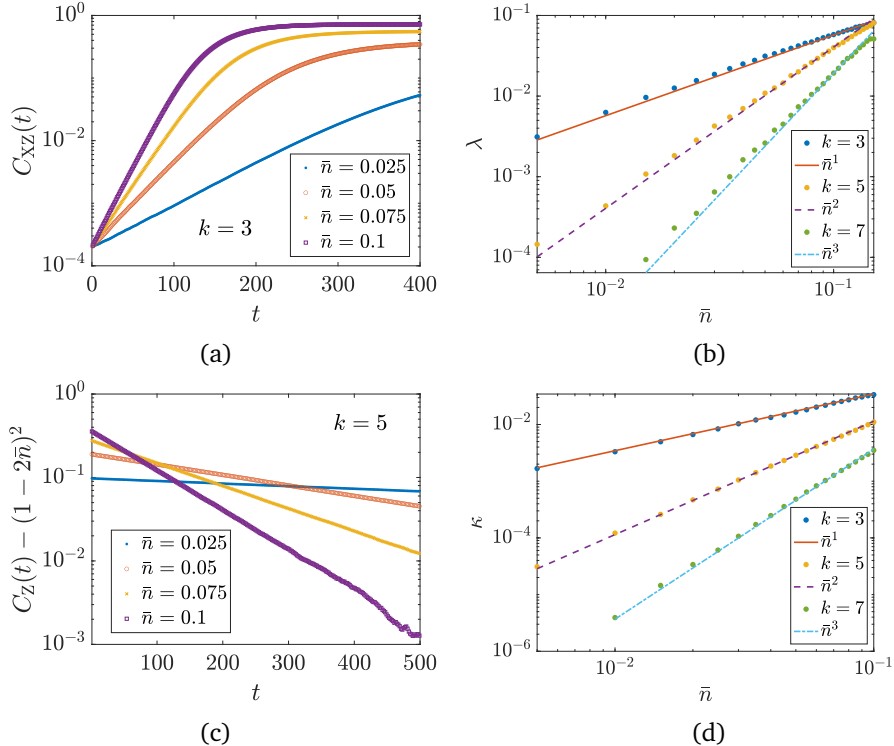

Figure 1: The correlation functions for random QA model with $N = 20000$ and $f = 0.5$. (a) The OTOC $C_{XZ}(t)$ vs time on the semi-log scale. (b) The Lyapunov exponent $\lambda$ vs $\bar{n}$ on the log-log scale for various $k$.(c) The auto correlator $C_Z(t)$ vs time on the semi-log scale. (d) The exponent $\kappa$ vs $\bar{n}$ on the log-log scale for various $k$.

## 5.2 Butterfly velocity

We have also studied the butterfly velocity $v_B$ in QA circuits where the degrees of freedom are arranged in a one-dimensional line [41]. The circuit for $k = 5$ is shown in Fig. 2; observe that the $U_k$ gates can now only act on a set of $k$ adjacent degrees of freedom on the line: $|s_{i+1}s_{i+2}\cdots s_{i+k}\rangle$. The QA circuits with $k = 3$ and $k = 7$ are constructed in an analogous way. In this case, there is no Lyapunov exponent due to the spatial locality. Nevertheless, we expect that

$$v_B \sim \bar{n}^{(k-1)/2}, \quad (\bar{n} \ll 1). \tag{5.89}$$

The time-dependent randomness ensures that the $\bar{n}$ exponent "derived" in (3.42) must be multiplied by a factor of 2. Numerically, we computed $v_B$ by performing data collapse of the front of $C_{XZ}(r,t)$ ( See the example in Fig. 3(a)). We confirmed this prediction, as shown in Fig. 3(b).

## 6 Conclusions

We derived a new bound (3.40) on the growth of operators (as measured by OTOCs in a suitable (grand) canonical ensemble) in arbitrary many-body quantum systems. We studied several large $N$ models with U(1) symmetry and showed that in the highly polarized sector with charge density $\bar{n} \ll 1$, the charged SYK model saturates our bound while the random dynamics including Brownian SYK model and random quantum automaton circuit do not. Due to the

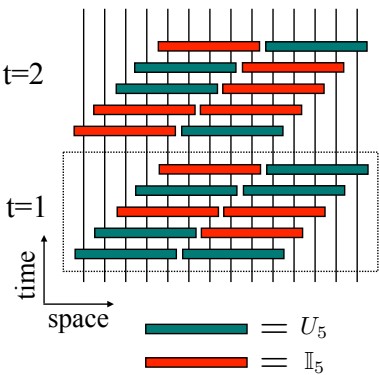

Figure 2: The local random QA circuit with 5-qubit gate. A single period of the circuit consists 5-layers. The block is a 5-qubit gate which randomly picks an identity operator or $U_5$ gate with equal probability. The dashed box indicates the circuit in one time step.

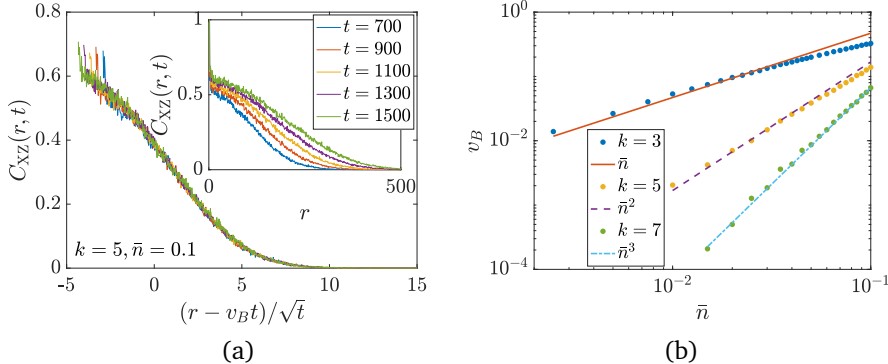

Figure 3: (a) The data collapse for the front of $C_{XZ}(r,t)$ with $k = 5$ and $L = 1000$. The curves at different time in the inset collapse into a single curve when we take $v_B = 0.1132$. (b) The butterfly velocity $v_B$ as a function of $\overline{n}$ for different $k$. Different from the model with all-to-all interaction in Sec. 5.1, we take $\overline{n} \equiv N^{\downarrow}/N$ for numerical convenience.

randomness in the time direction, the latter class of models lose the quantum coherence which allows the SYK model to saturate our bound. The Lyapunov exponents in these two classes of models satisfy the scaling relation

$$\lambda_{L,\text{quantum}}^2 \sim \lambda_{L,\text{classical}}, \tag{6.90}$$

and therefore classical systems are much less chaotic than quantum systems. Remarkably, a similar phenomenon to (6.90) arises in the study of systems with long-range interactions, where operator growth is much slower in effectively classical models [44,45] than in quantum coherent models [46–48].

There are a number of interesting applications and extensions of our work, which we briefly mention. Firstly, it is certainly interesting to try and generalize our results to other kinds of symmetry groups. An obvious candidate is SU(2) symmetry, which is easily realized in models of interacting qubits of the kind discussed in this paper. Such systems can approximately be realized in cold atomic gases [49], and our bounds may be relevant for designing models where highly entangled and metrologically useful states [50] exhibit very long lifetimes.

Secondly, we proposed a heuristic "bound" (3.42) on the butterfly velocity $v_B$, which characterizes the growth of operators in a many-body model on the lattice. It would be interesting to make that argument more rigorous, if possible. More interestingly, it is worth investigating whether or not the density dependence of the butterfly velocity is captured by (3.42), or by the random unitary circuit models, which predicts (for a fermionic model such as SYK)

$$v_B \sim \overline{n}^{(q-2)/2}. \tag{6.91}$$

We postulate that, as in [41], the scaling (6.91) is more robust, as it incorporates destructive interference effects that seem natural for a typical chaotic system.

Thirdly, recent work has used similar random circuits to model aspects of quantum gravity. Our work suggests that such analogies could be misleading for understanding short-time dynamics [21, 27, 35, 51], because the mechanism for the exponential OTOC growth (1.2) is subtly different in a random circuit versus a holographic model. It would be interesting to understand better the crossover between the quantum coherent operator growth in the SYK model, and quantum incoherent operator growth in a random circuit, in particular to better understand quantum dynamics in chaotic lattice models. We also comment that random circuits have more recently been used to model holographic questions on much longer time scales, including the dynamics of a large and evaporating black hole [52, 53]. Our work has no obvious relationship to this interesting problem.

Lastly, we note that other authors [54, 55] has recently obtained the following bound for charged systems at chemical potential $\mu$ and temperature $T$:

$$\lambda_L \leq \frac{2\pi T}{1 - |\mu|/\mu_c}, \tag{6.92}$$

where $\mu_c$ is a constant beyond which the (grand) canonical ensemble does not exist. We believe that this result, while it could be tight, is special to rotating black holes and their holographic duals. For example, the rotating three-dimensional black hole is dual to a two-dimensional conformal field theory with holomorphic factorization, in which case $T$ represents the harmonic mean of the left/right-moving temperatures (each of which controls a separate Lyapunov bound). In contrast, our result shows that (at least at infinite temperature) dynamics slows down by going to a constrained part of the Hilbert space. We expect that our results are much more universal, especially in non-holographic models. It would be interesting to generalize our result to finite temperature $T$, in which case a more detailed comparison with (6.92) could be made, along with other holographic results [56].

# Acknowledgements

We acknowledge Pengfei Zhang for useful discussions. AL is supported by a Research Fellowship from the Alfred P. Sloan Foundation. YG is supported by the Gordon and Betty Moore Foundation EPiQS Initiative through Grant GBMF-4306, and the US Department of Energy through Grant DE-SC0019030.

# A  Details for the regular and Brownian SYK calculations

In this appendix, we provide a few more details about our SYK calculation using the Keldysh formalism. As shown in Fig. 4, correlators are defined on a doubled Keldysh contour [57]. We introduce $(u, d)$ labels for each contour depending on whether time runs forwards or

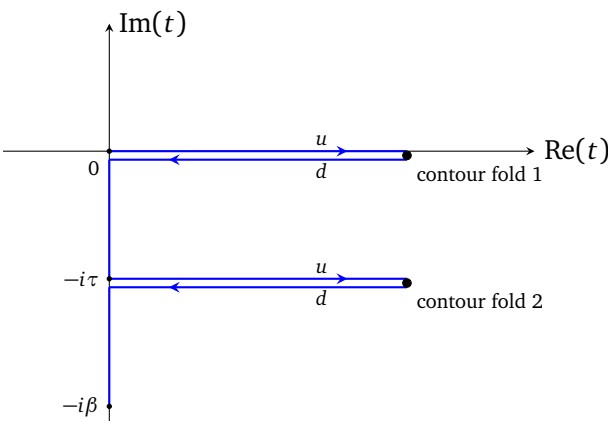

Figure 4: Keldysh contour with multiple contour folds. In this figure, we draw 2 contours, each fold consists of two sides (rails), upper ($u$) and lower ($d$) which are connected on the right end. We connect the different Keldysh contours on the left via imaginary time evolution, i.e. the state we start with is a thermal equilibrium.

backwards, and also introduce $\alpha = 1, 2 \ldots N$ for the contour indices. The interaction vertex is diagonal in $(u, d)$ basis, so it will be convenient to first express the self energy in the $(u, d)$ basis, and later make the basis change to the conventional $(K, R, A)$ as follows:

$$\begin{pmatrix} G^K & G^R \\ G^A & 0 \end{pmatrix} = \frac{1}{2} \begin{pmatrix} 1 & 1 \\ 1 & -1 \end{pmatrix} \begin{pmatrix} G^{uu} & G^{ud} \\ G^{du} & G^{dd} \end{pmatrix} \begin{pmatrix} 1 & 1 \\ 1 & -1 \end{pmatrix},$$

$$\begin{pmatrix} 0 & \Sigma^A \\ \Sigma^R & \Sigma^K \end{pmatrix} = \frac{1}{2} \begin{pmatrix} 1 & 1 \\ 1 & -1 \end{pmatrix} \begin{pmatrix} \Sigma^{uu} & \Sigma^{ud} \\ \Sigma^{du} & \Sigma^{dd} \end{pmatrix} \begin{pmatrix} 1 & 1 \\ 1 & -1 \end{pmatrix}. \tag{A.93}$$

For complex fermions, we need to be careful about the arrows when drawing diagrams. $G(t_1, t_2)$ is represented by an arrow from $t_2$ to $t_1$. We then find that

$$\Sigma_{\alpha\beta}^{ab}(t_1, t_2) = i \left( t_1, \alpha, a \begin{array}{c} \longleftarrow \\ \end{array} t_2, \beta, b \right).$$

$$= \pm i J^2 (i G_{\alpha\beta}^{ab}(t_1, t_2))(G^{ab}(t_1, t_2)_{\alpha\beta} G^{ba}(t_2, t_1)_{\beta\alpha})^{\frac{q-2}{2}}, \quad +(-) \quad \text{for} \quad \alpha \neq (=)\beta. \tag{A.94}$$

Here superscripts $a, b \in \{u, d\}$ label the rail, the $+$ sign is for $a \neq b$, and the $-$ sign for $a = b$. Subscripts $\alpha, \beta = 1 \ldots N$ label the contour index. The sign structure is due to the rule that each vertex is associated with a coupling constant: $-iJ$ for $u$ vertex, $+iJ$ for $d$ vertex.

Now we are ready to compute the self-energy

$$\Sigma^R = \frac{1}{2} \left( \Sigma^{uu} + \Sigma^{ud} - \Sigma^{du} - \Sigma^{dd} \right), \tag{A.95}$$

which is diagonal in contour index. To proceed, we use the quasi-particle form (4.53) to obtain the following Green's functions, in the limit $\beta \to 0$ with $\overline{\mu} = \beta\mu$ fixed:

$$G^{uu}(t) \approx -i \left( \Theta(t) \frac{e^{i\mu t - \Gamma|t|}}{1 + e^{\beta\mu}} - \Theta(-t) \frac{e^{i\mu t - \Gamma|t|}}{1 + e^{-\beta\mu}} \right), \quad G^{ud}(t) \approx i \frac{e^{i\mu t - \Gamma|t|}}{1 + e^{-\beta\mu}},$$

$$G^{du}(t) \approx -i \frac{e^{i\mu t - \Gamma|t|}}{1 + e^{\beta\mu}}, \quad G^{dd}(t) \approx -i \left( -\Theta(t) \frac{e^{i\mu t - \Gamma|t|}}{1 + e^{-\beta\mu}} + \Theta(-t) \frac{e^{i\mu t - \Gamma|t|}}{1 + e^{\beta\mu}} \right). \tag{A.96}$$

Note the useful combinations

$$G^{uu}(t) G^{uu}(-t) = G^{dd}(t) G^{dd}(-t) = G^{ud}(t) G^{du}(-t) \approx \frac{e^{-2\Gamma|t|}}{4 \cosh^2 \frac{\beta\mu}{2}}. \tag{A.97}$$

Finally we obtain (4.56) for the regular SYK model:

$$\Sigma^{\mathrm{R}}(t) = -\mathrm{i}\Theta(t)\frac{J^2}{(2\cosh\frac{\beta\mu}{2})^{q-2}}\mathrm{e}^{-(q-1)\Gamma t}\mathrm{e}^{\mathrm{i}\mu t} \qquad (\text{regular}). \tag{A.98}$$

Switching to Brownian SYK, we only need to change the coupling $J^2$ to $J\delta(t)$ and further simplify, i.e.

$$\Sigma^{\mathrm{R}}(t) = -\mathrm{i}\Theta(t)\frac{J\delta(t)}{(2\cosh\frac{\beta\mu}{2})^{q-2}}\mathrm{e}^{-(q-1)\Gamma t}\mathrm{e}^{\mathrm{i}\mu t} = -\mathrm{i}\Theta(t)\frac{J\delta(t)}{(2\cosh\frac{\beta\mu}{2})^{q-2}} \qquad (\text{Brownian}), \tag{A.99}$$

which is shown as (4.63) in the main text.

Now, we come to the contour index off-diagonal components. The superscripts $(u, d)$ do not matter any more; the ordering is determined by the subscripts completely. Thus,

$$\Sigma^{\mathrm{K}}_{21}(t) = -2\mathrm{i}J^2(\mathrm{i}G_{21}(t))(G_{21}(t)G_{12}(-t))^{\frac{q-2}{2}}, \tag{A.100}$$

where we consider 21 component (rather than 12) since contour 1 is customarily with smaller imaginary time, and we denote the imaginary time separation of two contours by $\tau$, i.e. $\psi_1(t) = \psi(t)$, $\psi_2(t) = \psi(t - \mathrm{i}\tau)$. One can also use the Keldysh function $G^{\mathrm{K}}$ instead of the plain one above, which differs by a factor of 2, namely $G^{\mathrm{K}}_{12} = 2G_{12}$, $G^{\mathrm{K}}_{21} = 2G_{21}$. Therefore

$$\Sigma^{\mathrm{K}}_{21}(t) = \frac{J^2}{2^{q-2}}(G^{\mathrm{K}}_{21}(t)G^{\mathrm{K}}_{12}(-t))^{\frac{q-2}{2}}G^{\mathrm{K}}_{21}(t). \tag{A.101}$$

Again, in the limit $\beta \to 0$ with fixed $\overline{\mu}$ and $\Gamma$, we have

$$G^{\mathrm{K}}_{21}(t) \approx -\mathrm{i}\frac{2\mathrm{e}^{\mu\tau}}{1 + \mathrm{e}^{\beta\mu}}\mathrm{e}^{\mathrm{i}\mu t - \Gamma|t|}, \quad G^{\mathrm{K}}_{12}(t) \approx \mathrm{i}\frac{2\mathrm{e}^{\mu(\beta-\tau)}}{1 + \mathrm{e}^{\beta\mu}}\mathrm{e}^{\mathrm{i}\mu t - \Gamma|t|}, \tag{A.102}$$

and the following product has a simple expression:

$$G^{\mathrm{K}}_{21}(t)G^{\mathrm{K}}_{12}(-t) \approx \frac{\mathrm{e}^{-2\Gamma|t|}}{(\cosh\frac{\beta\mu}{2})^2}. \tag{A.103}$$

Thus, the rung function

$$R(t) = \frac{\delta\Sigma^{\mathrm{K}}}{\delta G^{\mathrm{K}}} = (q-1)\frac{J^2}{2^{q-2}}\left(G^{\mathrm{K}}_{21}(t)G^{\mathrm{K}}_{12}(-t)\right)^{\frac{q-2}{2}} \approx (q-1)J^2\frac{\mathrm{e}^{-(q-2)\Gamma|t|}}{(2\cosh\frac{\beta\mu}{2})^{q-2}} \qquad (\text{regular}). \tag{A.104}$$

Similarly, switching to the rung function for Brownian SYK amounts to changing $J^2$ to $J\delta(t)$

$$R(t) = (q-1)\frac{J\delta(t)}{2^{q-2}}\left(G^{\mathrm{K}}_{21}(t)G^{\mathrm{K}}_{12}(-t)\right)^{\frac{q-2}{2}} = (q-1)J\delta(t)\frac{1}{(2\cosh\frac{\beta\mu}{2})^{q-2}} \qquad (\text{Brownian}). \tag{A.105}$$

The above two derivations explain the (4.59) and (4.66) in the main text.

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
