# Peer review of "Many-body quantum dynamics slows down at low density"

_SciPost Physics, doi:SciPost Phys. 9, 071 (2020)_

## Round 2 · Referee Report · Anonymous (Referee 1) · 2020-9-3

Strengths

  1. Clarity of presentation including all definitions
  2. Derivation of a sensible result using interesting methods

Report

This work derives bounds on the Lyapunov exponent in the low density limit of quantum systems with U(1) charge conservation. Given that bounds and scaling properties of Lyapunov exponents are hard to explicitly derive, this is a useful contribution. The work is methodical, and language and definitions are made clear. Some small points:

  1. "In this paper, we elect to study a simpler analogue of low temperature physics – a system at infinite temperature," Sounds like an oxymoron, so maybe the authors can unpack what they really mean by this statement?
  2. Page 3 reference is missing.
  3. typo "resultt" on page 7.

---

## Round 2 · Referee Report · Anonymous (Referee 2) · 2020-9-30

Strengths

  1. This is a nice discussion of a physically clear phenomenon: chaos is suppressed at low densities since there are fewer collisions.
  2. The inner-product based approach is elegant and fairly transparent, and leads to a simple derivation of density-dependent bounds.
  3. The calculations on SYK and Brownian SYK are clear.

Weaknesses

  1. I would have liked some cleaner physical motivation of the n^(1/2) dependence in the low density limit. It is clear that the operator size must behave as exp(-c mu) for some constant but this does not fix the exponent. In the limiting case it feels like some more elementary argument should be available and I encourage the authors to lay it out / find it.

Report

I think this paper is interesting and can be published.

---

## Round 3 · Referee Report · Anonymous (Referee 1) · 2020-11-3

Report

The author's have added clarifications as I had requested. I am satisfied and recommend for publication.

---

## Round 3 · Referee Report · Anonymous (Referee 2) · 2020-11-4

Report

I thank the authors for answering my question, and now recommend publication.

---

## Round 3 · Author Response

Dear editor/referees,

We thank you for the very positive reports. We have made the minor requested changes to our manuscript and believe it is now ready for publication.

Sincerely,
Xiao, Yingfei, Andy

---

## Round 3 · List of Changes

Response to the first referee:

We have fixed the typos that were pointed out, and modified the discussion above (1.4) to clarify what we meant by finite density at infinite T as an “analogue” of finite T: both correspond to dynamics in a very small part of Hilbert space.

Response to the second referee:

We added a new paragraph to the introduction (containing equations (1.6) and (1.7)) which explains why (1.5) is a universal bound at finite density.

Another tiny change:

In our discussion of the Schwinger-Keldysh formalism, we changed "world index" to "contour index" since the latter appears more customary in the literature.

---

## Editorial Decision

published